# Profiling Reduced Expression of Contractile and Mitochondrial mRNAs in the Human Sinoatrial Node vs. Right Atrium and Predicting Their Suppressed Expression by Transcription Factors and/or microRNAs

**DOI:** 10.3390/ijms251910402

**Published:** 2024-09-27

**Authors:** Weixuan Chen, Abimbola J. Aminu, Zeyuan Yin, Irem Karaesmen, Andrew J. Atkinson, Marcin Kuniewicz, Mateusz Holda, Jerzy Walocha, Filip Perde, Peter Molenaar, Halina Dobrzynski

**Affiliations:** 1Division of Cardiovascular Sciences, The University of Manchester, Manchester M13 9PL, UK; weixuan.chen@postgrad.manchester.ac.uk (W.C.); abimbola.aminu@yahoo.com (A.J.A.); zeyuan.yin@postgrad.manchester.ac.uk (Z.Y.); irem.karaesmen@postgrad.manchester.ac.uk (I.K.); andrew.atkinson-2@manchester.ac.uk (A.J.A.);; 2Department of Anatomy, Jagiellonian University Medical College, 31-008 Krakow, Poland; 3HEART-Heart Embryology and Anatomy Research Team, Department of Anatomy, Jagiellonian University Medical College, 31-034 Krakow, Poland; 4National Institute of Legal Medicine, 042122 Bucharest, Romania; filipvirgil@gmail.com; 5Northside Clinical School of Medicine, The University of Queensland, The Prince Charles Hospital, Brisbane, QLD 4072, Australia

**Keywords:** human sinus node/sinoatrial node, miRNA, transcription factor, contractile function, mitochondrial function, glycogen metabolism, Ingenuity Pathway Analysis, human right atrium

## Abstract

(1) Background: The sinus node (SN) is the main pacemaker of the heart. It is characterized by pacemaker cells that lack mitochondria and contractile elements. We investigated the possibility that transcription factors (TFs) and microRNAs (miRs) present in the SN can regulate gene expression that affects SN morphology and function. (2) Methods: From human next-generation sequencing data, a list of mRNAs that are expressed at lower levels in the SN compared with the right atrium (RA) was compiled. The mRNAs were then classified into contractile, mitochondrial or glycogen mRNAs using bioinformatic software, RStudio and Ingenuity Pathway Analysis. The mRNAs were combined with TFs and miRs to predict their interactions. (3) Results: From a compilation of the 1357 mRNAs, 280 contractile mRNAs and 198 mitochondrial mRNAs were identified to be expressed at lower levels in the SN compared with RA. TFs and miRs were shown to interact with contractile and mitochondrial function-related mRNAs. (4) Conclusions: In human SN, TFs (MYCN, SOX2, NUPR1 and PRDM16) mainly regulate mitochondrial mRNAs (COX5A, SLC25A11 and NDUFA8), while miRs (miR-153-3p, miR-654-5p, miR-10a-5p and miR-215-5p) mainly regulate contractile mRNAs (RYR2, CAMK2A and PRKAR1A). TF and miR-mRNA interactions provide a further understanding of the complex molecular makeup of the SN and potential therapeutic targets for cardiovascular treatments.

## 1. Introduction

The sinus node (SN) is a crescent-shaped structure located at the junctional area of the superior vena cava, inferior vena cava and crista terminalis in the posterior wall of the right atrium (RA) (refer to Figure 1a) [1,2]. The SN structure is embedded in an extensive amount of connective tissue, and the pacemaker cells in the SN are characterized by their small size and empty appearance due to the absence of mitochondria and contractile machinery. By inference, the SN is not a contractile tissue, unlike the surrounding RA [3,4,5]. Furthermore, the SN does not have deposits of β-particles of glycogen [6,7]. The presence of fewer glycogen granules differentiates it from RA, which can be visualized to identify the SN in micro-computed tomography scans, as shown in Figure 1a [1].

Transcription factors (TF) are crucial transcriptional activators that regulate gene expression, including the regulation of transcription during embryogenesis [8]. It has been shown previously that TFs can regulate the expression of ion channels and Ca^2+^-handling proteins, which modulate SN function. The T-box TF TBX3 is expressed in both developing and matured SN in mammals [3,9,10]. The SHOX2 TF also plays a role in sinus venosus development, and SHOX2-deficient mice and zebrafish show bradycardia [10,11,12].

MicroRNA (miR) are small non-coding RNAs with short sequences that regulate post-transcription by inhibiting the gene expression [13]. miRs bind primarily to the 3′ untranslated regions of the target mRNA to promote mRNA degradation or translational repression [14,15,16]. They play a role in the development of various cardiovascular diseases, such as coronary artery disease, arrhythmias, and heart failure [16]. miRa can affect SN morphology and function. Our group previously showed that the overexpression of miR-486-3p down-regulated the expression of HCN4, a key pacemaker gene in the SN, to cause bradycardia [17].

Contractile function plays a crucial part in maintaining cardiac function, and a process called excitation–contraction coupling enables cardiac chamber contraction and relaxation from the electrical excitation of the myocyte. Ca^2+^ enters the cell through depolarization during the action potential, which triggers the sarcoplasmic reticulum (SR) to release more Ca^2+^ into the cytosol via the ryanodine receptor (RYR2). The increased concentration of intracellular Ca^2+^ allows the binding of Ca^2+^ with the myofilament protein troponin C, leading to the activation of contractile machinery [18,19]. miRs and TFs can impact contractile function. For example, the absence of miR-208 has been shown to decrease the mechanism of β-myosin heavy chain expression in response to pressure overload, thus reducing contractility in response to increased pressure [20]. miR-25 suppresses the expression of SERCA2, which regulates the calcium decay by transporting Ca^2+^ from the cytosol into the sarcoplasmic reticulum to cause relaxation of the myocyte [21]. The TF Vezf1 regulates the β-myosin heavy chain, and then regulates the cardiac contractile function and links with dilated cardiomyopathy [22].

Mitochondrial function is important for maintaining cardiac contractility. The contraction–relaxation cycle requires a large amount of energy, and the majority of the energy is derived from mitochondrial oxidative phosphorylation, which is a process that synthesizes energy-rich phosphate bonds into adenosine triphosphate [23,24]. Mitochondrial dysfunction can lead to cardiovascular disease, such as hypertrophy, diabetic cardiomyopathy, and heart failure with preserved ejection fraction [25,26,27]. It has been shown that miRs are present in the mitochondrion, and different concentrations of miRs are linked with various cardiovascular diseases [28,29,30]. For example, miR-181c targets the cytochrome c oxidase subunit 1 (COX1) to regulate the mitochondrial genome and is linked with heart failure [31]. TFs are also involved in regulating mitochondrial function in the heart. For instance, overexpression of the mitochondrial transcription factor A (TFAM) increases the amount of mitochondrial DNA, leading to myocardial infarction and resulting in reduced cardiac function [32]. T-box transcription factor 20 (TBX20) strengthens both the contractile and mitochondrial functions of the heart [33].

In this study, mRNAs that are related to contractile function, mitochondrial function and glycogen metabolism were identified to help us better understand the SN molecular makeup and function. Both TFs and miRs can inhibit or suppress protein expression. Since their effect on the SN is unknown, their interactions were investigated in the present study. A better understanding of their effects on protein expression in the SN could lead to novel therapeutic targets to treat SN dysfunction and/or atrial fibrillation.

## 2. Results

### 2.1. Differential mRNA Expression in the Human Sinus Node and Atrial Muscle

As published previously by our group, the next-generation sequencing data that compared mRNA levels in human SN and RA were attained [12]. A total of 3060 mRNAs were identified, and 1357 out of these 3060 mRNAs were identified to show significantly lower expression in the SN compared to RA, including mRNAs that were later classified as relevant for contractile function, including RYR2, CAMK2A, PRKAR1A, and mRNAs that were later classified as relevant for mitochondrial function, including COX5A, SLC25A11 and NDUFA8 (Figure 1b). These 1357 mRNAs would be further classified into contractile function, mitochondrial function and glycogen metabolism by using gene ontology, as shown in Figure 2.

In Figure 2, after gene ontology analysis, the mRNAs were classified into three categories, biological process, cellular compartment and molecular function. The cellular compartment contains the most mRNAs (242 mRNAs) followed by the molecular function (193 mRNAs) and the biological process (145 mRNAs). Combining the mRNAs from three categories, 238 mRNAs were identified to relate to contractile function and 170 mRNAs were identified to relate to mitochondrial function. The mRNAs that were expressed at lower levels in the SN were further classified into different canonical pathways by using Ingenuity Pathway Analysis (IPA; Qiagen, Germany) in Figure 3.

From IPA in Figure 3, we see that a total of 27 canonical pathways were identified to show the greatest negative z-score, indicating the contribution of these pathways to lower mRNA expression in the SN compared to RA. Among these pathways, the citric acid (TCA) Cycle II, calcium signaling and oxidative phosphorylation pathways showed the most negative expression in the SN compared to RA. The TCA Cycle II and oxidative phosphorylation pathways relate to mitochondrial function, and the calcium signaling pathway closely relates to contractile function.

A total of 121 mRNAs were identified in these canonical pathways, of which 42 mRNAs were involved with contractile function and 28 mRNAs were involved with mitochondrial function. A summary of the canonical pathways and the names of the mRNAs involved in each pathway are shown in Table 1.

Eight mRNAs in the dataset are shown to have a relationship with glycogen metabolic pathways. Five of them were shown in later analysis to have direct interactions with TFs and miRs. The mean counts of these five mRNAs are shown in Figure 1c.

### 2.2. Differential TFs and miRs Expression in the Human SN and Atrial Muscle

In previous publications from our group, 68 TFs and 18 miRs were identified to have a significantly higher expression in the SN compared to the RA (Figure 4) [12,17]. The next step was to determine how these TFs and miRs interact with and regulate the mRNAs shown in Figure 1, Figure 2 and Figure 3.

### 2.3. Interactions between mRNAs and miRNAs, TFs

There are complex interactions between the mRNAs involved with contractile function, mitochondrial function, glycogen metabolism, TFs and miRs. The mRNAs that were predicted to have direct interactions with TFs or miRs are summarized in Table 2. A total of 55 contractile function-related mRNAs, 52 mitochondrial function-related mRNAs, and 5 glycogen metabolism-related mRNAs are shown.

Figure 5 shows 25 TFs that have interactions with 62 mRNAs. Among those mRNAs, 24 mRNAs are involved with contractile function and 38 mRNAs are involved with mitochondrial function. The TFs that are expressed at higher levels in SN compared to RA are more related to mRNAs that are involved with mitochondrial function than mRNAs that are involved with contractile function. MYCN, SOX2, NUPR1 and PRDM16 have the highest numbers of interactions with the mRNAs that are related to mitochondrial function, such as COX5A (links with MYCN and PRDM16), SLC25A11 (links with SOX2, MYCN, PRDM16, VAV1 and CEBPA) and NDUFA8 (links with MYCN and LMO2). The detailed predicted interactions between TFs and mRNAs are summarized in Table 3.

Figure 6 shows 16 miRs that have interactions with 62 mRNAs. Among the mRNAs, 42 mRNAs are involved with contractile function, and 20 mRNAs are involved with mitochondrial function. The miRNAs that are more highly expressed in the SN than in the RA are more related to mRNAs that are involved with contractile function than mRNAs that are involved with mitochondrial function.

The predicted number of binding sites can be found in Table 4, which is arranged from the most interactions to the least interactions. Only the interactions that are predicted to have binding sites are shown in the table. miR-153-3p, miR-654-5p, miR-10a-5p and miR-215-5p link with the highest number of contraction-related mRNAs, such as RYR2 (links with miR-153-3p and miR-198), CAMK2A (links with miR-654-5p), CAMK2B (links with miR-10a-5p, miR-1225-3p, miR-512-5p), and PRKAR1A (links with miR-215-5p).

We next determined which mRNAs are regulated by both TF and miR and the network for TFs, miRs and mRNAs. mRNAs that have interactions with both TFs and miRs are summarized in Figure 7. There were 12 contraction-related mRNAs and 6 mitochondria-related mRNAs that interact with both TFs and miRNAs. For example, PRKAA2 has interactions with two miRNAs and three TFs, PRKACA has interactions with three miRNAs and one TF, PRKAR1A has interactions with one miRNA and three TFs, and COX5A has interactions with one miR and two TFs.

From the mRNA dataset, a total of eight mRNAs were identified to relate to glycogen metabolism, which could help to explain the attenuation difference between the SN region and the surrounding RA that was observed in micro-CT scans (refer to Figure 1a,c) [1]. From Figure 8, we see that five out of eight glycogen metabolic mRNAs are predicted to have direct interactions with TFs and miRs. SLC2A4 links with one miR and two TFs, and ALDOA links with two miRNAs and two TFs.

## 3. Discussion

From this study, mRNAs that have lower expression levels in the SN than the RA were determined, analyzed and categorized. mRNAs that are relevant to contractile mechanisms, mitochondrial function and glycogen metabolism were identified. It was found that mRNAs with lower expression levels in the SN compared with RA had interactions with TFs and miRs that had higher expression levels in the SN than in RA (Figure 5, Figure 6, Figure 7 and Figure 8).

The contractile and mitochondrial functions are important in maintaining normal SN and atrial function. Contractile and/or mitochondrial dysfunction have been linked with SN dysfunction and atrial fibrillation [34,35]. Although relatively “empty” in the SN, mitochondria still exist in a small number, and a further decrease in mitochondria disrupts the mitochondria–SR microdomain and leads to SN dysfunction [36]. Then, SN dysfunction further triggers atrial fibrillation [34].

Some mRNAs related to contractile function, such as RYR2, CAMK2A and PRKAR1A, and mitochondrial function, such as COX5A, SLC25A11, and NDUFA8, were identified. The RYR2 gene encodes the cardiac ryanodine receptor, a Ca^2+^ handling protein. RYR2 mutations are responsible for catecholaminergic polymorphic ventricular tachycardia and exon 3 deletion syndrome. Fifty-eight per cent of patients with exon 3 deletion syndrome have sinus node dysfunction [37,38]. This shows the correlation of RYR2 expression with maintaining the SN function. Our group has previously demonstrated that RYR2 is expressed at lower levels in human SN compared to RA [12,39]. In this study, RYR2 was classified as a contractile function-related mRNA, which had interactions with two miRs. We observed that NDUFA8 was associated with mitochondrial function, consistent with another study. NDUFA8 was down-regulated in SARS-CoV-2-infected cardiomyocytes, resulting in mitochondrial dysfunction and cardiomyocyte apoptosis. NDUFA8 is associated with mitochondrial function, consistent with what is shown in this study, and down-regulated when infected with SARS-CoV-2 [40].

It was found that transcription factors are more likely to be relevant for mRNAs that are involved in mitochondrial function than mRNAs related to contractile function.

Transcription factors MYCN, SOX2, NUPR1, and PRDM16 were shown to interact with many mRNAs that specifically link with mitochondrial function. MYCN has interactions with 29 mRNAs, and 25 of them are related to mitochondrial function. This protein is located in the nucleus, where it associates with DNA binding activities, essential for apoptosis and proliferation. Myocardial MYCN plays an essential role in maintaining ventricular wall morphogenesis, and loss results in reduced cardiomyocyte proliferation and hypocellular myocardium [41]. As shown in Figure 8, MYCN also has interactions with ALDOA, a glycogen metabolic-related mRNA. SOX2 has interactions with 14 mRNAs and 9 mRNAs related to mitochondrial function. SOX2 was shown to be associated with BMP and WNT pathway activity. The activation of these pathways promotes the cardiac induction of human embryonic stem cells [42]. NUPR1 has interactions with seven mRNAs, and five of them are related to mitochondrial function. NUPR1 is located in the nucleus and regulates many cellular processes, including DNA repair, ER stress, and apoptosis [43,44]. PRDM16 is another transcription factor that has interactions with five mRNAs, and four of them are related to mitochondrial function. PRDM16 has been shown to play an important role in cardiomyocytes, and the loss of PRDM16 causes dilated cardiomyopathy and left ventricular noncompaction cardiomyopathy [45,46,47]. Various research studies suggest functions of these four TFs (MYCN, SOX2, NUPR1 and PRDM16) in the cardiovascular system; however, the specific functions of the above four TFs in SN have not been shown previously. The significantly high expression of these TFs in SN compared with RA and the large number of predicted interactions with the mRNAs suggest an important role for TFs in maintaining SN morphology and function.

We determined that miRNAs are more likely to be related to mRNAs that are concerned with contractile function than mitochondrial function. miR-153-3p is one of the more important miRNAs, which has 19 interactions with mRNAs, with 16 of these being contraction-related mRNAs. It has been shown that miR-153-3p plays an important role in mitochondrial fission and cardiomyocyte hypertrophy by inhibiting the expression of the Mfn1 protein [48]. However, in our bioinformatic analysis, miRs showed even more importance in regulating the expression of contractile-related mRNAs, which indicates a potential therapeutic target for treating cardiovascular diseases, especially cardiac hypertrophy.

miR-654-5p has 15 interactions with mRNAs, and 11 of these interactions are with the contractile-related mRNAs. In previous research, miR-654-5p was shown to be involved with cell migration and proliferation, where it maintains normal cardiovascular structures, especially artery structures.

miR-10a-5p has 13 interactions with mRNAs, and 10 of these mRNAs are related to contractile function. miR-10a-5p has been shown to play an essential role in regulating and maintaining endothelial function in athero-susceptible regions and inhibiting the progression of atherosclerosis by promoting mitochondrial fatty acid oxidation [49,50].

miR-215-5p has nine interactions with mRNAs, and seven of these mRNAs are involved in contractile function. In recent research carried out by Latimer et al., an increased level of miR-215-5p has been observed in cardiomyocyte circadian clock-disrupted mice [51], meaning this miRNA may play a critical role in maintaining cardiac physiology.

In general, microRNAs play an important role in regulating heart function, such as cardiac growth and conduction [52]. However, their interactions with various mRNAs and the impacts thereof have not been deeply studied, especially the impact on contractile machinery in the cardiovascular system. miRs may provide potential novel therapeutic targets for cardiac diseases.

Two mRNAs, PRKAA2 and PRKAR1A, have been identified to have the most interactions with both miRNAs and TFs. PRKAA2 is a catalytic subunit of the AMP-activated protein kinase (AMPK), and it is a cardiac-specific enhancer that is predominantly active in the myocardium [53]. PRKAR1A is one isoform of the protein kinase A (PKA) regulatory subunit, and PKA is one of the key regulators in the excitation-contraction coupling [54]. PRKAR1A inhibition has been shown to suppress cardiomyocyte hypertrophy through mitochondrial fission [55]. It is known that AMPK interplays with PKA to regulate oxidative stress and mitochondrial function [56]. These examples show the close relationship of these mRNAs with mitochondrial function, and indicate the potential targets for SN research.

In this study, the predicted networks between mRNA and TF/miR were explored specifically in human SN. This allows a better understanding of the main regulators and their impacts on the morphology and function of human SN, and provides new therapeutic targets for various cardiovascular disease treatments.

## 4. Materials and Methods

### 4.1. Next-Generation Sequencing for mRNA, Bioinformatic Analysis for TFs and qPCR for miRs

In this study, only the datasets from human samples were used. Next-generation sequencing was performed at the Genomic Technologies Core Facility at the University of Manchester. Three SN and three RA samples used for next-generation sequencing (NGS) experiments were then used for mRNA expression. A HiSeq4000 instrument (Chicago, IL, USA) was used for paired-end sequencing and mRNA quantification. Bcl2fastq software (2.17.1.14) was used to generate the mRNA expression database. The individual values from three SNs and three RAs, mean values of the SN and the RA, Log2fold change, *p* values, adjusted *p* values and calculated percent difference from the SN and RA specimens were produced and provided by the Core Facility. More details can be found in previous publications from our group [12,17]. From a total of 3060 mRNAs identified, 1357 mRNAs that were significantly (*p* < 0.05) expressed at lower levels in SN vs. RA were used for further investigation.

To determine which and how TFs and miRs interact with mRNAs to inhibit or suppress the mRNAs that are expressed at lower levels in the SN, the TFs and miRs that were expressed at lower levels in the SN vs. RA were investigated further in this study. In a previous study conducted by Aminu et al., TFs were identified using the bioinformatic software, Ingenuity Pathways Analysis (IPA version 01-22-01; Qiagen, Germany). In that study, a dataset of 68 TFs that were significantly more highly expressed in the SN vs. RA with the Log2Foldchange cut-off of 0 to 1.5 was selected and used [12].

Seven SN and seven RA samples were used for qPCR for miR expression as described in Petkova et al. (2020) and Aminu et al. (2021) [12,17]. From a total of 66 miRs that showed significant differences in their expression in the human SN vs. RA, the dataset of 18 miRs that were more significantly highly expressed in the SN vs. RA was used in this study.

### 4.2. Bioinformatic Analysis to Identify the mRNAs That Relate to Contractile Function, Mitochondrial Function and Glycogen Metabolism

#### 4.2.1. RStudio Analysis

The Bioconductor version 3.16 was used in RStudio software (link to Bioconductor: https://bioconductor.org/news/bioc_3_16_release/ [bioconductor.org]). The “ClusterProfiler” package was used (link to the package: https://bioconductor.org/packages/release/bioc/html/clusterProfiler.html [bioconductor.org]). The “enrichGO” command and human gene library in the package were used (inputs: enrichGO (gene = genes_to_test, OrgDb = “org.Hs.eg.db”, keyType = “ENSEMBL”, ont = “BP” or “CC” or “MF”)). The 1357 mRNAs, as mentioned in Section 2.1, (the gene IDs of which can also be obtained from the corresponding author upon request), that are significantly expressed at lower levels in SN over RA were used as the gene list for the GO results to categorize these mRNAs into “biological process (BP)”, “cell component (CC)”, and “molecular function (MF)” (Figure 2). The sub-categories of the BP, CC and MF belong to mitochondrial function and contractile function, except the categories indicated with the red arrows in Figure 2. The gene IDs of the mRNAs that were involved with each sub-category were identified and used for further analysis.

#### 4.2.2. Canonical Pathway Analysis in IPA

All 1357 mRNAs that showed lower expression in SN were uploaded into the software Ingenuity Pathways Analysis (IPA version 01-22-01; Qiagen, Germany). The uploaded information included gene ID, base mean values, Log2fold change, *p* values and adjusted *p* values. The “core analysis” was used and the “canonical pathway” tab was selected to show the top canonical pathways that have a significantly low expression in the SN compared to RA. The mRNAs involved with each canonical pathway were identified (Figure 3).

#### 4.2.3. Contractile Function, Mitochondrial Function, and Glycogen Metabolism Identification

As indicated in Section 4.2.1, the sub-categories of the BP, CC and MF were identified using the RStudio, and gene IDs of the mRNAs involved with each sub-category were identified (Figure 2). By using online human gene databases, such as Genome (https://genome.ucsc.edu), NIH National Library of Medicine (https://www.ncbi.nlm.nih.gov), GeneCards (https://www.genecards.org) and MedlinePlus (https://medlineplus.gov), these mRNAs were assessed for contractile function, mitochondrial function, and glycogen metabolism. Here, 238 mRNAs were identified that were relevant to contractile function, and 170 mRNAs were related to mitochondrial function via RStudio analysis. As indicated in Section 4.2.2, mRNAs that were involved in the canonical pathways were identified in IPA (Figure 3). Using the same online human genes databases, the mRNAs identified from IPA were also assessed for contractile function, mitochondrial function, and glycogen metabolism. Here, 42 mRNAs were relevant for contractile function and 28 mRNAs for mitochondrial function. In total, eight mRNAs were identified that were relevant for glycogen metabolism.

### 4.3. Creating and Analysing mRNA, TF and miR Interactions/Networks in IPA

All mRNAs that were relevant to contractile function, mitochondrial function and glycogen metabolism were uploaded into IPA using the “path designer” function in the software to build the interactions among these mRNAs, TFs and miRs. The “path explorer” in the “path designer” function was used, and only the “direct” interactions were selected.

These mRNAs were first combined with the 68 TFs on IPA using the “path designer” function by putting the names of the mRNAs and TFs into the software. Only the predicted direct interactions are shown in the figure (Figure 5). Then, the mRNAs were combined with 18 miRs by putting the names of the mRNAs and miRs into the software, and the direct inhibitions are shown in the figure (Figure 6). Finally, as TFs and miRs can all act on mRNAs to regulate gene expression, mRNAs, TFs and miRs were combined by putting the names of mRNAs, TFs and miRs together into the software. Only the mRNAs that have direct interactions with both TFs and miRs were left on the figure to find out the interactions among three players (Figure 7). In total, eight glycogen metabolism-related mRNAs were also combined with TFs and miRs, as shown in Figure 8.

### 4.4. Binding Site Prediction between mRNAs and miRs

For each predicted interaction between mRNA and miR that we retrieved from IPA, we further confirmed the number of binding sites. The sequences of mRNAs and miRs were obtained by putting the names of the mRNA into Genome (https://genome.ucsc.edu/index.html) and putting the names of the miRs into miRBase (https://www.mirbase.org) accordingly. Then, RNA22 (https://cm.jefferson.edu/rna22/Interactive/) and TargetScanHuman (https://www.targetscan.org/vert_72/) were used to predict the number of binding sites from the sequences of the mRNAs and miRs. The number of binding sites is recorded in Table 4. The predicted interaction between mRNA and miR is shown in Figure 6, Figure 7 and Figure 8. The interactions that show no binding sites from RNA22 and TargetScanHuman confirmations were deleted from the interaction/network map (Figure 6, Figure 7 and Figure 8).

### 4.5. Heat Maps, Graphs, and Statistical Analysis

Heatmaps were created using Heatmapper (http://www.heatmapper.ca/expression/). An Excel spreadsheet that contains gene names and the normalized count of each gene expressed at significantly lower levels in each SN sample (*n* = 3) over the RA sample (*n* = 3) was created and uploaded onto Heatmapper (Figure 1b). Greener represents a higher while redder represents a lower expression in the SN vs. RA. A Z-score of −1 indicates that a sample is one standard deviation below the mean value, and a Z-score of 1 indicates a sample is one standard deviation above the mean value (Figure 1b).

GraphPad Prism 10.0.3. and Microsoft Excel were used for statistical analysis and for making graphs of mean counts of glycogen metabolism mRNAs in Figure 1c, while *p* values and adjusted *p* values of TFs and miRs are shown in Figure 4.

## 5. Limitations

One limitation of this study is the lack of validation of predicted interactions between mRNAs and TFs, and interactions between mRNAs and miRs. To validate the interactions between mRNAs and TFs, chromatin immunoprecipitation (ChIP) assays can be used to discover novel genes that can interact with specific transcription factors, as previously reported [57]. To validate the interactions between mRNAs and miRs, luciferase reporter gene assays can be used to validate the inhibition of mRNAs at the 3′UTR region by miRs, as previously reported by our group [17,58].

It should also be noted that during dissection, we only dissected the main SN body and the RA from the pectinate muscles; however, other cell types, such as the endothelial cells, fibroblasts and macrophages, are also present in the samples, instead of purely pacemaker cells from the SN and cardiomyocytes from the RA.

## 6. Conclusions

In the human SN, the mitochondrial function-related mRNAs, such as COX5A, SLC25A11, and NDUFA8, are mainly regulated by TFs, such as MYCN, SOX2, NUPR1 and PRDM16, while contractile function-related mRNAs, such as RYR2, CAMK2A and PRKAR1A, are mainly regulated by miRs, such as miR-153-3p, miR-654-5p, miR-10a-5p and miR-215-5p. Some AMPK and PKA-related mRNAs, such as PRKAA2 and PRKAR1A, are highly involved with both TF and miR interactions. This classification of mRNAs can provide a better understanding of the SN’s molecular makeup, and the TF and miR interactions can provide potential therapeutic targets for the treatment of cardiovascular diseases. 

## Figures and Tables

**Figure 1 ijms-25-10402-f001:**
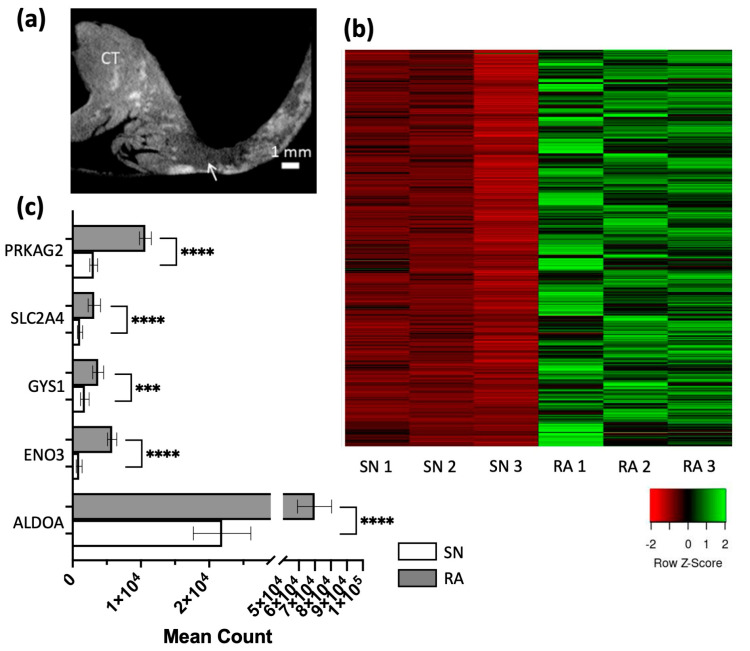
(**a**) Micro-computed tomography (micro-CT) scan of the human SN region (as previously shown and identified by Stephenson et al., 2017 [1]). The white arrow points to the darker region, which is identified as the SN region. (**b**) Heatmap of the 1357 mRNAs that are expressed at lower levels in the SN than RA. Greener color indicates higher gene expression and redder color indicates lower gene expression. (**c**) mRNAs relevant to glycogen metabolism with lower gene expression in SN compared to RA. The names of these mRNAs are listed on the *Y*-axis. The adjusted *p* values between SN and RA are shown for each mRNA, *** 0.0001 < *p* ≤ 0.001, **** *p* < 0.0001.

**Figure 2 ijms-25-10402-f002:**
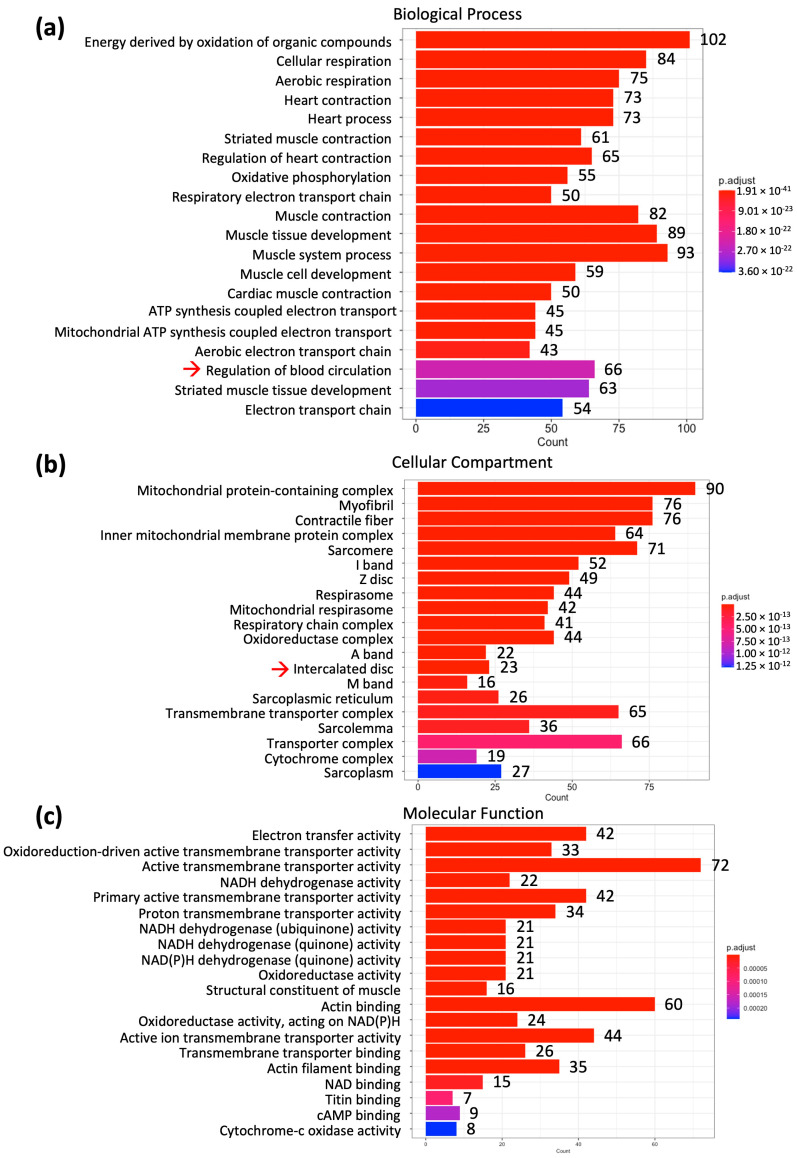
Gene ontology analysis performed to categorize 1357 mRNAs expressed at lower levels in SN compared to RA. The red arrows point to the categories that are not related to contractile function, mitochondrial function, or glycogen metabolism, and therefore the genes in these categories were deleted from further analysis. (**a**) Biological process category of the mRNAs. (**b**) Cellular compartment category of the mRNAs. (**c**) Molecular function category of the mRNAs.

**Figure 3 ijms-25-10402-f003:**
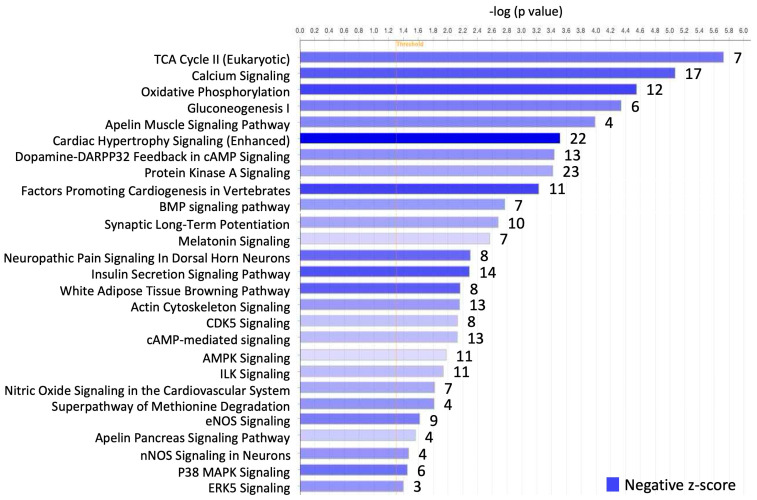
Ingenuity Pathway Analysis shows the significantly lower-expressed canonical pathways in the SN compared to RA. The numbers on the right of each bar indicate the numbers of genes involved in each canonical pathway. The blue colour represents the negative z-score.

**Figure 4 ijms-25-10402-f004:**
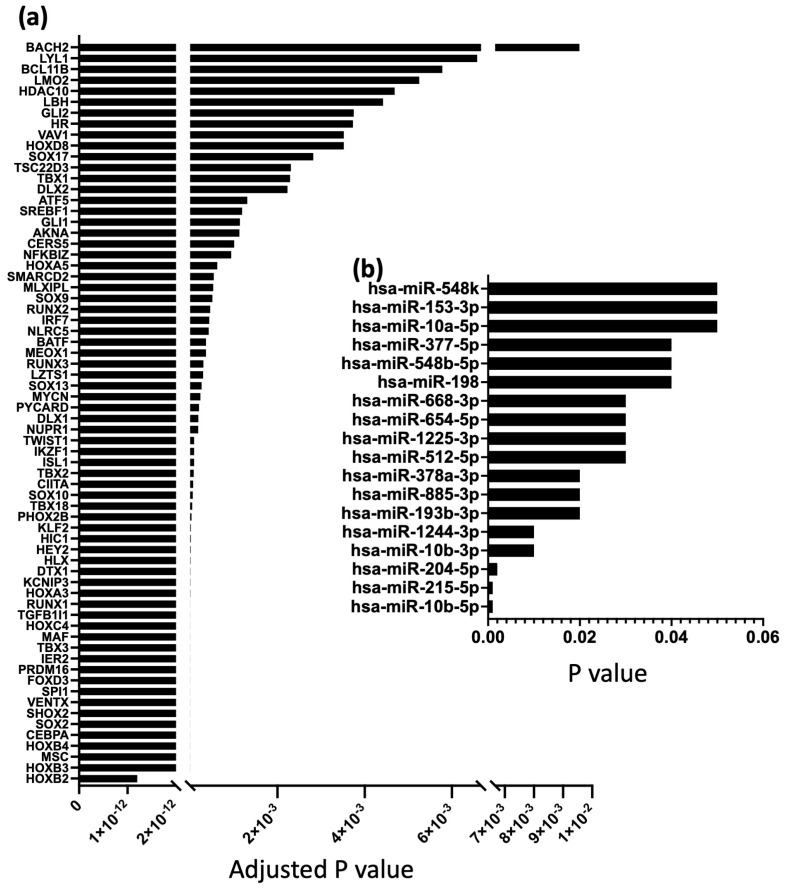
Adjusted *p* values of significantly more highly expressed transcription factors (**a**) and miRNAs (**b**) in SN compared to RA.

**Figure 5 ijms-25-10402-f005:**
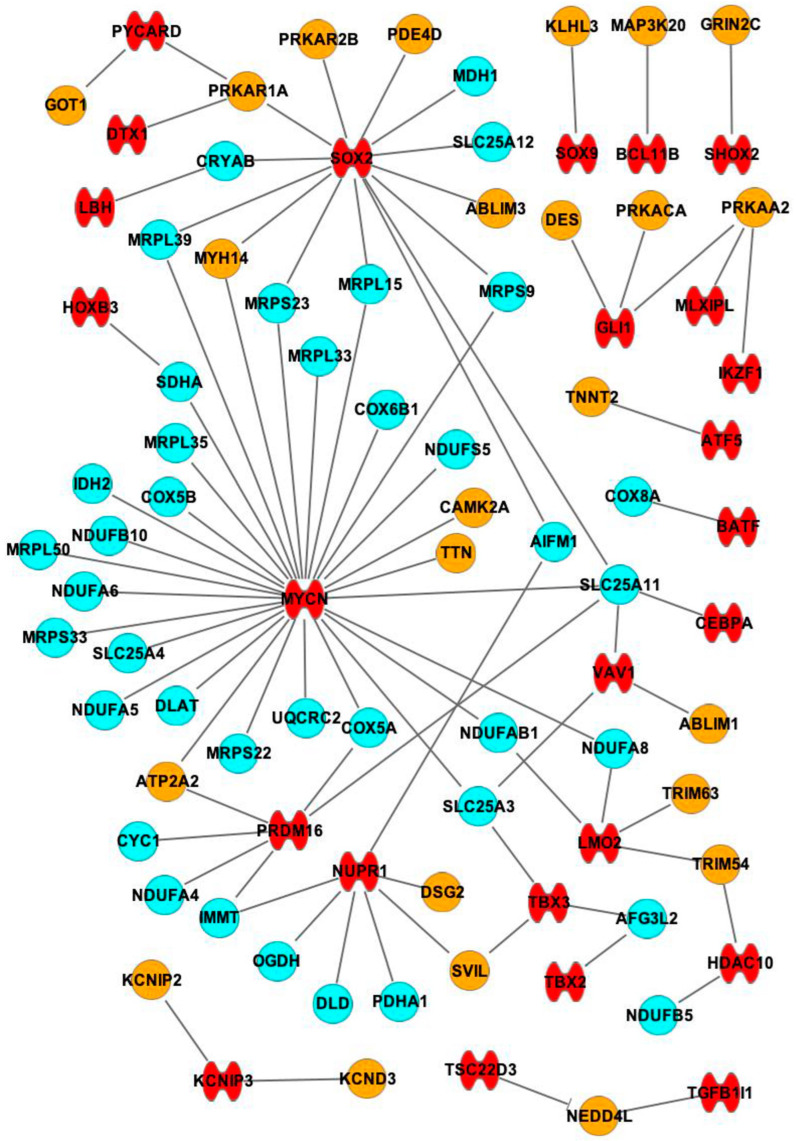
Interactions of the higher-expressed TFs and the lower-expressed mRNAs in the SN compared with RA. A summary of the predicted interactions between TFs and mRNAs is listed in Table 3. Blue: mRNAs that are involved with mitochondrial function. Orange: mRNAs that are involved with contractile function. Red: TFs. —: interaction. —|: inhibition.

**Figure 6 ijms-25-10402-f006:**
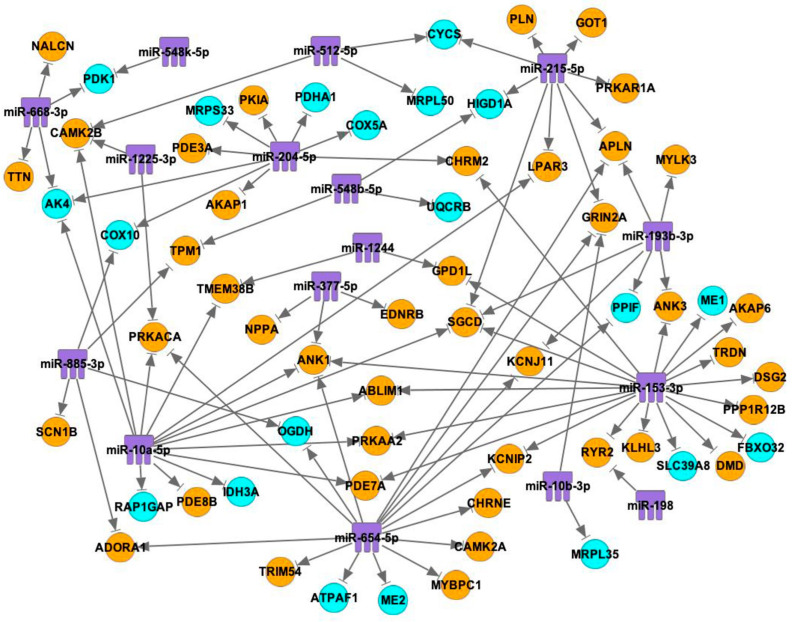
Interactions of the more highly expressed miRs and lower-expressed mRNAs in SN compared to RA. A summary of the predicted interactions between miRs and mRNAs is listed in Table 4 Blue: mRNAs that are involved in mitochondrial function. Orange: mRNAs that are involved in contractile function. Purple: miRs. →|: inhibition.

**Figure 7 ijms-25-10402-f007:**
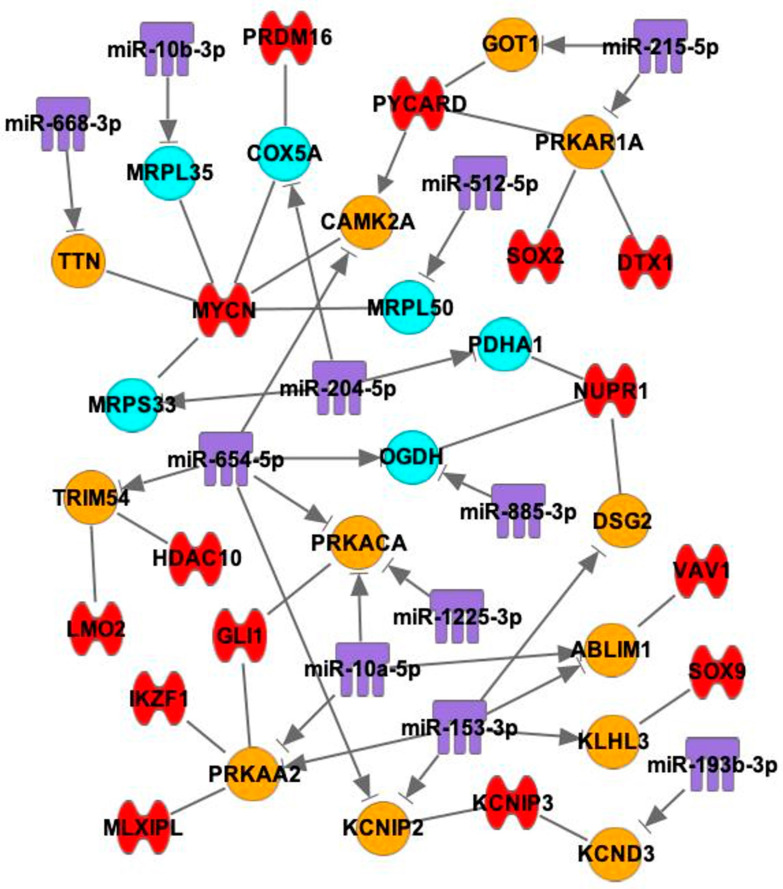
mRNAs that have predicted interactions with both TFs and miRs are shown. A summary of the predicted interactions with TFs, miRNAs and mRNAs is listed in Table 2 and Table 3. Blue: mRNAs that are involved with mitochondrial function. Orange: mRNAs that are involved with contractile function. Red: TFs. Purple: miRs. —: interaction. →|: inhibition.

**Figure 8 ijms-25-10402-f008:**
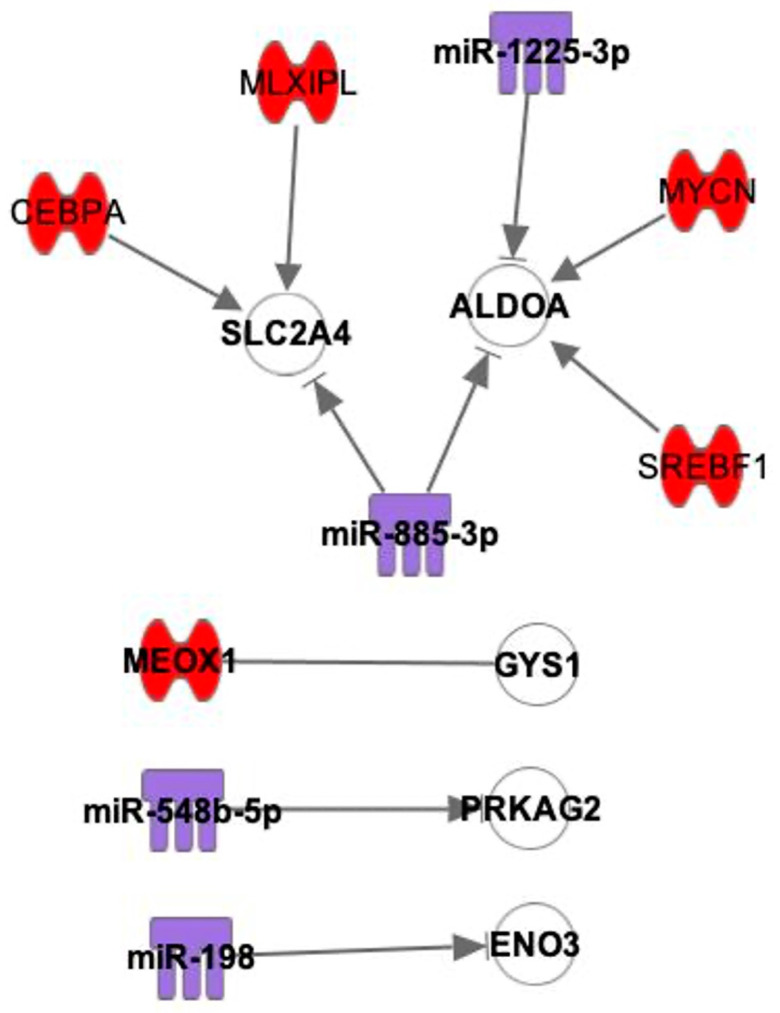
Interactions of mRNAs involved with glycogen metabolism with TFs and miRs. The expression levels of the glycogen metabolic mRNAs are shown in Figure 1c. Red: TFs. Purple: miRs. White: mRNAs that are involved with glycogen metabolism. —: interaction. →|: inhibition.

**Table 1 ijms-25-10402-t001:** List of canonical pathways that IPA predicted to contribute to low mRNA expression in SN vs. RA, and the list of genes involved in each pathway (also refer to Figure 3). The table is arranged from the least to greatest −log *p* values, and shows pathway names with the number of genes involved and gene names.

Pathway (Number of Genes Involved)	Gene
TCA Cycle II (Eukaryotic) (7)	ACO2, CS, FH, IDH3A, MDH1, OGDH, SDHA
Calcium Signaling (17)	ATP2A2, CAMK2A, CAMK2B, CASQ2, CHRNE, GRIN2A, GRIN2C, MYH6, MYH14, MYL4, PNCK, PRKAG2, PRKAR2B, RCAN2, RYR2, TNNT2, TRDN
Oxidative Phosphorylation (12)	ATPAF1, COX10, COX5A, COX6A2, CYCS, NDUFA4, NDUFA5, NDUFAB1, NDUFS1, SDHA, UQCRB, UQCRC2
Gluconeogenesis I (6)	ALDOA, ENO3, MDH1, ME1, ME2, PGAM2
Apelin Muscle Signaling Pathway (4)	APLN, PRKAA2, PRKAG2, SLC2A4
Cardiac Hypertrophy Signaling (Enhanced) (22)	ATP2A2, CAMK2A, CAMK2B, HSPB3, HSPB7, IFNLR1, IL5RA, MAP3K9, MAP3K20, MAPKAPK3, NPPA, PDE3A, PDE7A, PDE8B, PDK1, PLCL1, PLN, PRKAG2, PRKAR2B, RASD1, RPS6KA5, RYR2
Dopamine-DARPP32 Feedback in cAMP Signaling (13)	ATP2A2, GRIN2A, GRIN2C, KCNJ11, PLCL1, PPM1L, PPP1R3A, PPP1R3C, PPP2R2C, PPP2R3A, PRKAG2, PRKAR2B, PRKG2
Protein Kinase A Signaling (23)	AKAP1, AKAP3, AKAP6, CAMK2A, CAMK2B, EYA1, MYL4, MYLK3, NTN1, PDE3A, PDE7A, PDE8B, PHKG1, PLCL1, PLN, PPP1R3A, PPP1R3C, PRKAG2, PRKAR2B, PTPRQ, PYGM, RYR2, TTN
Factors Promoting Cardiogenesis in Vertebrates (11)	BMP7, BMP10, BMP8A, CAMK2A, CAMK2B, GJA5, MYH6, NPPA, PLCL1, SCN5A, TBX5
BMP signaling pathway (7)	BMP7, BMP8A, BMP10, PRKAG2, PRKAR2B, RASD1, SOSTDC1
Synaptic Long Term Potentiation (10)	CAMK2A, CAMK2B, GRIN2A, GRIN2C, PLCL1, PPP1R3A, PPP1R3C, PRKAG2, PRKAR2B, RASD1
Melatonin Signaling (7)	CAMK2A, CAMK2B, PLCL1, PRKAG2, PRKAR2B, RORC, SLC2A4
Neuropathic Pain Signaling In Dorsal Horn Neurons (8)	CAMK2A, CAMK2B, GRIN2A, GRIN2C, KCNH2, PLCL1, PRKAG2, PRKAR2B
Insulin Secretion Signaling Pathway (14)	CAMK2A, CAMK2B, DLAT, GIPR, KCNJ11, NALCN, PDHA1, PLCL1, PRKAG2, PRKAR2B, RPS6KA5, RYR2, SLC2A4, STAT4
White Adipose Tissue Browning Pathway (8)	BMP7, FNDC5, NPPA, PRKAA2, PRKAG2, PRKAR2B, PRKG2, VEGFA
Actin Cytoskeleton Signaling (13)	ACTN2, EGF, FGF11, FGF12, MYH6, MYH14, MYL4, MYLK3, PAK6, PIP5K1B, PPP1R12B, RASD1, TTN
CDK5 Signaling (8)	PPM1L, PPP1R3A, PPP1R3C, PPP2R2C, PPP2R3A, PRKAG2, PRKAR2B, RASD1
cAMP-mediated signaling (13)	ADORA1, AKAP1, AKAP3, AKAP6, CAMK2A, CAMK2B, CHRM2, PDE3A, PDE7A, PDE8B, PKIA, PRKAR2B, RAP1GAP
AMPK Signaling (11)	AK4, CHRM2, CHRNE, CKM, PPM1L, PPP2R2C, PPP2R3A, PRKAA2, PRKAG2, PRKAR2B, SLC2A4
ILK Signaling (11)	ACTN2, DSP, ITGB6, MYH6, MYH14, MYL4, PPM1L, PPP2R2C, PPP2R3A, RPS6KA5, VEGFA
Nitric Oxide Signaling in the Cardiovascular System (7)	ATP2A2, PLN, PRKAG2, PRKAR2B, PRKG2, RYR2, VEGFA
Superpathway of Methionine Degradation (4)	GOT1, GOT2, PCCB, PRMT8
eNOS Signaling (9)	AQP4, AQP7, CHRM2, CHRNE, LPAR3, PRKAA2, PRKAG2, PRKAR2B, VEGFA
Apelin Pancreas Signaling Pathway (4)	APLN, PRKAA2, PRKAG2, PRKAR2B
nNOS Signaling in Neurons (4)	CAMK2A, GRIN2A, GRIN2C, RASD1
P38 MAPK Signaling (6)	HSPB3, HSPB7, MAPKAPK3, PLA2G12B, RPS6KA5, TIFA
ERK5 Signaling (3)	RASD1, RPS6KA5, WNK1

**Table 2 ijms-25-10402-t002:** List of mRNAs predicted to have direct interactions with TFs and miRs expressed at higher levels in the SN compared to RA. The table is arranged from the least to the greatest adjusted *p* values, and shows mRNA names, SN mean ± SEM; RA mean ± SEM; Log2fold change; and adjusted *p* value. Black = mRNAs that are related to glycogen metabolism. Blue: mRNAs that are related to mitochondrial function. Orange: mRNAs that are related to contractile function.

mRNA	SN Mean ± SEM	RA Mean ± SEM	Log2fold Change	Adjusted *p* Value
NPPA	40,366.13 ± 11,840.28	587,003.64 ± 233,459.53	−3.80	3.17 × 10^−47^
GRIN2C	150.49 ± 41.54	1505.33 ± 206.64	−3.40	1.32 × 10^−29^
GRIN2A	34.41 ± 9.15	395.65 ± 63.35	−3.61	1.20 × 10^−21^
LPAR3	239.30 ± 78.64	1563.00 ± 250.50	−2.87	2.03 × 10^−19^
ENO3	947.30 ± 433.60	5786.00 ± 672.30	−2.89	9.77 × 10^−18^
TMEM38B	435.93 ± 66.81	2180.97 ± 137.96	−2.37	5.96 × 10^−17^
FBXO32	1823.69 ± 622.47	8064.94 ± 2141.13	−2.22	3.57 × 10^−16^
RYR2	6201.72 ± 2293.58	43,858.17 ± 21,127.23	−2.77	1.82 × 10^−15^
CHRNE	532.36 ± 146.12	2493.47 ± 302.70	−2.33	2.01 × 10^−14^
MYLK3	2917.00 ± 966.30	16,247.00 ± 1172.00	−2.65	5.87 × 10^−14^
PDE3A	1669.00 ± 133.00	6510.00 ± 506.40	−1.98	9.75 × 10^−14^
ME2	1040.19 ± 75.53	4288.91 ± 471.40	−2.05	1.07 × 10^−13^
PPP1R12B	8186.14 ± 2305.97	37,247.20 ± 6630.09	−2.25	6.54 × 10^−13^
RAP1GAP	107.50 ± 12.52	544.70 ± 21.50	−2.37	7.62 × 10^−13^
KCNIP2	1871.96 ± 431.95	9718.33 ± 735.63	−2.47	1.86 × 10^−12^
SCN1B	458.62 ± 60.01	1785.74 ± 293.19	−1.96	4.18 × 10^−12^
PDE8B	506.40 ± 113.30	1798.00 ± 188.20	−1.91	2.64 × 10^−10^
ATP2A2	32,967.84 ± 8525.27	140,042.55 ± 11,888.16	−2.20	3.15 × 10^−10^
SGCD	1118.50 ± 130.93	4005.43 ± 752.98	−1.83	4.79 × 10^−10^
ANK1	326.65 ± 72.56	1302.83 ± 51.25	−2.09	5.94 × 10^−10^
PDK1	435.31 ± 85.02	1691.11 ± 137.84	−2.02	1.09 × 10^−9^
NEDD4L	548.57 ± 131.29	2313.96 ± 493.04	−2.13	1.11 × 10^−9^
UQCRB	8496.29 ± 1355.26	31,581.55 ± 7034.10	−1.85	1.68 × 10^−9^
PKIA	957.30 ± 178.40	3342.00 ± 294.50	−1.85	2.51 × 10^−9^
MYBPC1	80.81 ± 14.79	424.38 ± 67.97	−2.41	4.76 × 10^−9^
TRIM63	1263.61 ± 398.03	5687.17 ± 1969.60	−2.19	5.19 × 10^−9^
PRKAA2	952.70 ± 259.80	4127.00 ± 1218.00	−2.16	5.44 × 10^−9^
ME1	645.39 ± 142.07	2306.13 ± 46.86	−1.92	6.80 × 10^−9^
SLC39A8	726.23 ± 149.47	2476.79 ± 221.81	−1.83	1.51 × 10^−8^
PRKAG2	3090.00 ± 572.40	106,57.00 ± 858.00	−1.84	1.52 × 10^−8^
MYH14	1172.51 ± 230.82	3694.07 ± 614.29	−1.69	1.72 × 10^−8^
DSG2	1180.31 ± 381.92	4273.57 ± 852.77	−2.03	2.55 × 10^−8^
DMD	2411.16 ± 277.11	8679.83 ± 1901.66	−1.81	4.42 × 10^−8^
PRKAR2B	791.19 ± 108.73	2418.93 ± 79.89	−1.65	4.55 × 10^−8^
NDUFA5	2768.78 ± 345.70	8071.65 ± 861.85	−1.56	5.13 × 10^−8^
DLAT	1198.44 ± 194.96	3902.53 ± 187.84	−1.75	5.18 × 10^−8^
TTN	27,820.20 ± 8702.96	154,512.67 ± 72,399.52	−2.38	5.39 × 10^−8^
CHRM2	1265.95 ± 525.92	3877.67 ± 1095.15	−1.83	6.68 × 10^−8^
GOT1	4504.00 ± 1167.00	15,507.00 ± 642.50	−1.90	7.20 × 10^−8^
MAP3K20	5446.00 ± 1082.00	16,464.00 ± 3505.00	−1.60	9.15 × 10^−8^
AKAP1	1824.00 ± 512.30	5030.00 ± 656.60	−1.56	1.23 × 10^−7^
TRDN	2415.45 ± 701.59	10,011.69 ± 2499.98	−2.15	1.65 × 10^−7^
ABLIM3	2310.55 ± 372.12	6165.94 ± 1169.24	−1.42	1.95 × 10^−7^
KCNJ11	90.50 ± 26.67	370.59 ± 97.45	−2.04	2.16 × 10^−7^
CYCS	4494.10 ± 1257.61	14,511.32 ± 1868.14	−1.78	3.26 × 10^−7^
IDH3A	2512.14 ± 569.16	7355.81 ± 423.98	−1.63	5.48 × 10^−7^
SDHA	4783.13 ± 765.73	14,099.54 ± 1017.44	−1.60	5.76 × 10^−7^
CAMK2B	886.45 ± 242.81	2844.18 ± 389.22	−1.81	6.27 × 10^−7^
ALDOA	21,888.38 ± 4209.53	69,812.69 ± 10,672.80	−1.71	6.44 × 10^−7^
COX10	370.51 ± 82.92	1177.80 ± 41.61	−1.76	9.40 × 10^−7^
COX5A	3850.64 ± 932.14	12,170.01 ± 2045.61	−1.70	2.30 × 10^−6^
OGDH	4144.09 ± 905.90	11,254.98 ± 1322.87	−1.51	2.43 × 10^−6^
ATPAF1	2336.00 ± 496.90	6454.00 ± 15.82	−1.55	2.71 × 10^−6^
NALCN	200.50 ± 33.82	580.40 ± 38.73	−1.58	2.88 × 10^−6^
PDE7A	454.50 ± 93.88	1775.00 ± 711.30	−1.80	2.91 × 10^−6^
ABLIM1	8707.20 ± 1976.33	23,835.54 ± 8276.75	−1.39	3.61 × 10^−6^
PDHA1	4045.38 ± 917.17	11,743.03 ± 669.07	−1.63	4.49 × 10^−6^
NDUFA4	11,401.81 ± 2311.16	34,441.74 ± 5244.19	−1.63	5.67 × 10^−6^
PLN	13,987.76 ± 3665.84	42,146.81 ± 4579.06	−1.70	5.89 × 10^−6^
SLC25A4	14,584.36 ± 4389.42	46,121.40 ± 4482.91	−1.80	5.98 × 10^−6^
UQCRC2	5516.37 ± 839.18	15,418.20 ± 1675.86	−1.51	8.95 × 10^−6^
MDH1	9211.00 ± 2270.00	28,563.00 ± 4373.00	−1.70	1.04 × 10^−5^
APLN	53.12 ± 20.70	193.86 ± 52.80	−2.08	1.16 × 10^−5^
AK4	954.20 ± 245.80	2598.00 ± 608.00	−1.51	1.52 × 10^−5^
TRIM54	1293.40 ± 455.02	3632.34 ± 413.85	−1.75	1.71 × 10^−5^
AKAP6	1628.06 ± 423.62	4406.32 ± 259.99	−1.57	1.72 × 10^−5^
ANK3	2195.19 ± 135.79	5698.77 ± 854.85	−1.36	2.24 × 10^−5^
IDH2	2813.86 ± 680.68	7679.94 ± 545.31	−1.54	2.29 × 10^−5^
SLC2A4	1083.62 ± 376.68	3169.26 ± 913.33	−1.63	2.40 × 10^−5^
ADORA1	202.32 ± 75.96	563.38 ± 53.83	−1.72	3.06 × 10^−5^
NDUFAB1	2853.73 ± 666.19	8491.52 ± 1971.77	−1.56	3.21 × 10^−5^
IMMT	2548.38 ± 457.59	6293.75 ± 181.62	−1.37	3.55 × 10^−5^
PRKACA	3651.82 ± 592.69	8568.10 ± 302.59	−1.28	3.68 × 10^−5^
AIFM1	1250.79 ± 272.32	3128.84 ± 75.60	−1.41	6.67 × 10^−5^
TNNT2	52,996.68 ± 15,892.94	144,599.19 ± 21,446.60	−1.56	1.10 × 10^−4^
SLC25A12	1658.49 ± 208.14	3801.41 ± 191.81	−1.23	1.12 × 10^−4^
KLHL3	583.44 ± 155.63	1617.75 ± 772.65	−1.28	1.14 × 10^−4^
TPM1	59,923.00 ± 20,212.99	153,480.87 ± 31,542.18	−1.45	1.26 × 10^−4^
MRPL35	1054.26 ± 163.38	2374.22 ± 25.60	−1.22	1.46 × 10^−4^
KCND3	409.54 ± 78.90	1311.74 ± 614.24	−1.42	1.97 × 10^−4^
GPD1L	4931.18 ± 1450.70	10,273.33 ± 1593.37	−1.19	2.06 × 10^−4^
NDUFA8	1784.11 ± 388.01	4248.67 ± 601.70	−1.30	2.44 × 10^−4^
NDUFB5	2470.69 ± 347.08	5857.23 ± 754.88	−1.26	2.71 × 10^−4^
PDE4D	934.41 ± 155.71	2037.22 ± 476.12	−1.09	2.87 × 10^−4^
PRKAR1A	7776.36 ± 550.02	15,506.23 ± 1274.24	−1.01	3.71 × 10^−4^
CYC1	3470.36 ± 625.10	8433.44 ± 1428.91	−1.29	4.35 × 10^−4^
EDNRB	1215.00 ± 114.40	2583.00 ± 176.70	−1.11	4.69 × 10^−4^
GYS1	1783.00 ± 630.40	3749.00 ± 805.40	−1.21	5.40 × 10^−4^
MRPS23	674.46 ± 103.97	1396.07 ± 128.12	−1.08	5.40 × 10^−4^
HIGD1A	1351.36 ± 282.95	2904.10 ± 537.75	−1.12	5.62 × 10^−4^
MRPL33	1310.67 ± 180.02	2891.31 ± 306.91	−1.16	5.85 × 10^−4^
SLC25A11	1854.97 ± 336.97	4194.34 ± 509.14	−1.21	9.34 × 10^−4^
MRPL39	449.21 ± 36.86	985.84 ± 149.48	−1.12	9.68 × 10^−4^
NDUFA6	2675.74 ± 698.20	6039.42 ± 1259.69	−1.20	1.04 × 10^−3^
COX5B	6125.33 ± 1447.09	14,888.12 ± 3592.31	−1.26	1.10 × 10^−3^
DLD	2161.55 ± 418.65	4964.17 ± 623.10	−1.25	1.20 × 10^−3^
MRPL50	540.20 ± 47.38	1071.58 ± 32.52	−1.01	1.23 × 10^−3^
COX8A	1709.46 ± 377.55	4097.35 ± 964.90	−1.23	1.29 × 10^−3^
AFG3L2	1471.45 ± 191.94	2892.52 ± 40.73	−1.01	1.30 × 10^−3^
CAMK2A	226.40 ± 89.79	539.50 ± 31.36	−1.54	1.43 × 10^−3^
COX6B1	4041.13 ± 1030.85	9143.47 ± 2151.20	−1.18	1.55 × 10^−3^
MRPS22	683.43 ± 37.07	1432.12 ± 201.58	−1.05	1.57 × 10^−3^
NDUFB10	3759.25 ± 798.24	8448.57 ± 1735.75	−1.16	1.73 × 10^−3^
SLC25A3	10,661.69 ± 1643.67	22,876.10 ± 3179.36	−1.12	2.08 × 10^−3^
NDUFS5	4811.13 ± 932.38	9909.52 ± 2177.33	−1.02	2.92 × 10^−3^
PPIF	1692.49 ± 512.50	3511.49 ± 61.05	−1.24	3.34 × 10^−3^
MRPS33	721.80 ± 49.51	1461.57 ± 210.66	−1.01	3.43 × 10^−3^
CRYAB	42,557.08 ± 15,964.14	84,832.79 ± 15,603.49	−1.13	3.44 × 10^−3^
SVIL	6089.43 ± 1387.26	12,895.94 ± 3926.73	−1.04	3.67 × 10^−3^
MRPS9	865.43 ± 123.52	1804.79 ± 260.93	−1.07	4.47 × 10^−3^
DES	112,042.00 ± 40,572.00	202,808.00 ± 17,832.00	−1.14	7.18 × 10^−3^
MRPL15	1129.54 ± 234.79	2254.08 ± 289.96	−1.04	7.61 × 10^−3^

**Table 3 ijms-25-10402-t003:** Predicted interactions between TFs and mRNAs. The table shows TF names with the number of interactions related to each TF and the mRNA names of the predicted targets. TF: transcription factor. Blue: mRNAs that are related to mitochondrial function. Orange: mRNAs that are related to contractile function.

TF (Number of Interactions)	mRNA Target
MYCN (29)	MRPL15, MRPL33, MRPS23, MRPS9, MRPL39, SDHA, MRPL35, NDUFS5, COX5B, COX5A, IDH2, NDUFB10, MRPL50, NDUFA6, MRPS33, SLC25A4, NDUFA5, DLAT, SLC25A11, MRPS22, UQCRC2, COX6B1, SLC25A3, NDUFAB1, NDUFA8, TTN, ATP2A2, CAMK2A, MYH14
SOX2 (14)	SLC25A12, MDH1, CRYAB, SLC25A11, MRPL39, MRPS9, MRPS23, MRPL15, AIFM1, ABLIM3, MYH14, PRKAR2B, PRKAR1A, PDE4D
NUPR1 (7)	IMMT, DLD, PDHA1, OGDH, AIFM1, DSG2, SVIL
PRDM16 (5)	CYC1, IMMT, NDUFA4, SLC25A11, ATP2A2
LMO2 (4)	NDUFA8, NDUFAB1, TRIM54, TRIM63
GLI1 (3)	DES, PRKAA2, PRKACA
TBX3 (3)	SLC25A3, AFG3L2, SVIL
VAV1 (3)	SLC25A11, SLC25A3, ABLIM1
HDAC10 (2)	NDUFB5, TRIM54
KCNIP3 (2)	KCNIP2, KCND3
PYCARD (2)	GOT1, PRKAR1A
CEBPA (1)	SLC25A11
IKZF1 (1)	PRKAA2
TGFB1I1 (1)	NEDD4L
ATF5 (1)	TNNT2
BCL11B (1)	MAP3K20
BATF (1)	COX8A
DTX1 (1)	PRKAR1A
HOXB3 (1)	SDHA
LBH (1)	CRYAB
MLXIPL (1)	PRKAA2
SHOX2 (1)	GRIN2C
SOX9 (1)	KLHL3
TBX2 (1)	AFG3L2
TSC22D3 (1)	NEDD4L

**Table 4 ijms-25-10402-t004:** Predicted interactions between miRs and mRNAs. The table shows the miRNA name (number of interactions with mRNAs), mRNA names of the predicted target, and the number of binding sites for each miRNA on the predicted target created using RNA22 and TargetScanHuman. Blue: mRNAs that are related to mitochondrial function. Orange: mRNAs that are related to contractile function.

miRNA (Number of Interactions)	mRNA of Predicted Target	Number of Binding Sites for Each miRNA on Predicted Target (RNA22)	Number of Binding Sites for Each miRNA on Predicted Target (TargetScanHuman)
hsa-miR-153-3p (19)	ANK3	0	2
	TRDN	0	1
	DSG2	1	1
	PPP1R12B	0	1
	DMD	0	1
	KLHL3	1	3
	RYR2	0	2
	KCNIP2	0	1
	PDE7A	0	2
	PRKAA2	1	1
	SGCD	1	1
	AKAP6	0	1
	ABLIM1	1	1
	ANK1	0	1
	CHRM2	0	1
	GPD1L	0	1
	FBXO32	1	1
	SLC39A8	0	1
	ME1	1	1
hsa-miR-654-5p (15)	TRIM54	1	0
	MYBPC1	1	0
	CAMK2A	3	0
	CHRNE	1	0
	KCNIP2	1	0
	KCNJ11	2	0
	APLN	5	0
	GRIN2A	8	0
	ADORA1	1	0
	ANK1	2	0
	PRKACA	1	0
	OGDH	4	0
	ATPAF1	4	0
	ME2	4	0
	PPIF	4	0
hsa-miR-10a-5p (13)	PRKAA2	0	2
	PDE7A	0	2
	PDE8B	0	3
	CAMK2B	0	2
	TMEM38B	0	4
	PRKACA	0	1
	LPAR3	0	2
	ANK1	0	2
	ABLIM1	0	1
	SGCD	0	1
	IDH3A	0	1
	RAP1GAP	4	1
	AK4	0	7
hsa-miR-204-5p (9)	AKAP1	1	2
	CHRM2	0	7
	PDE3A	1	2
	PKIA	0	2
	MRPS33	2	0
	AK4	2	4
	COX10	0	1
	PDHA1	0	1
	COX5A	0	2
hsa-miR-215-5p (9)	PLN	0	2
	LPAR3	0	2
	GRIN2A	2	2
	SGCD	0	2
	PRKAR1A	0	2
	APLN	0	1
	GOT1	0	2
	HIGD1A	0	2
	CYCS	1	3
hsa-miR-193b-3p (6)	MYLK3	0	3
	ANK3	0	2
	APLN	0	1
	KCNJ11	0	1
	SGCD	0	2
	PPIF	0	2
hsa-miR-885-3p (5)	SCN1B	3	0
	TPM1	2	0
	ADORA1	5	0
	COX10	3	0
	OGDH	3	0
hsa-miR-668-3p (4)	TTN	1	1
	NALCN	2	0
	PDK1	4	0
	AK4	4	1
hsa-miR-548b-5p (3)	TPM1	1	0
	HIGD1A	2	0
	UQCRB	10	0
hsa-miR-512-5p (3)	CAMK2B	2	0
	CYCS	2	0
	MRPL50	1	0
hsa-miR-377-5p (3)	EDNRB	1	0
	ANK1	3	0
	NPPA	1	0
hsa-miR-1225-3p (2)	CAMK2B	2	0
	PRKACA	2	0
hsa-miR-1244 (2)	GPD1L	1	0
	TMEM38B	1	0
hsa-miR-10b-3P (2)	GRIN2A	2	0
	MRPL35	1	0
hsa-miR-198 (1)	RYR2	1	0
hsa-miR-548k (1)	PDK1	2	0

## Data Availability

All mRNA expression profiles are available from the corresponding author upon request.

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
