# Peer review of "Profiling Reduced Expression of Contractile and Mitochondrial mRNAs in the Human Sinoatrial Node vs. Right Atrium and Predicting Their Suppressed Expression by Transcription Factors and/or microRNAs"

_ijms, 2024, doi:10.3390/ijms251910402_

Round 1

Reviewer 1 Report

Comments and Suggestions for Authors

I suggest to reformulate the title in a more clear manner. It is quite difficult to understand the subsequent idea…

The study is a very interesting, very complex and exhaustive starting from the ideea that mRNAs is related to contractile function, mitochondrial function and glycogen metabolism and also miRs and transcription factors (TFs) can impact contractile function.

 Contractile function plays a crucial part in maintaining cardiac function, and the process excitation-contraction coupling enables cardiac contraction and relaxation triggered by the electrical excitation of the myocyte.

 Both TFs and miRs can inhibit or suppress protein expression. Since their effect on the SN is unknown, the present study tried to investigate this problem. A better understanding of their effects on protein expression in the SN could lead to novel therapeutic targets to treat SN dysfunction.

The introduction provides sufficient theoretical information about the importance of the research to be presented later.

But, for a better understanting of the study, I suggest that the material and method section should be presented and respectively moved before the results section .

In the abstract material and methods are presented at point no 2 before 3 (the results)

Reference no 5 is very old and with no direct relation with the manuscript.

 Some of my comments are inserted directly into the text.

Many phrases must be reformulated (from English language point of view)

In Disscusion section I suggest to present separately the limitations of the study.

Comments on the Quality of English Language

many phrases must be reformulated from english language point of view

Author Response

Thank you so much for carefully reviewing our paper. We appreciate your valuable time and effort. We have carefully addressed your concerns below.

Comments 1: I suggest to reformulate the title in a more clear manner. It is quite difficult to understand the subsequent idea…

Response 1: We have edited the title (highlighted in blue) to make it clearer.

Comments 2: The study is a very interesting, very complex and exhaustive starting from the ideea that mRNAs is related to contractile function, mitochondrial function and glycogen metabolism and also miRs and transcription factors (TFs) can impact contractile function.

Contractile function plays a crucial part in maintaining cardiac function, and the process excitation-contraction coupling enables cardiac contraction and relaxation triggered by the electrical excitation of the myocyte.

Both TFs and miRs can inhibit or suppress protein expression. Since their effect on the SN is unknown, the present study tried to investigate this problem. A better understanding of their effects on protein expression in the SN could lead to novel therapeutic targets to treat SN dysfunction.

The introduction provides sufficient theoretical information about the importance of the research to be presented later.

 But, for a better understanting of the study, I suggest that the material and method section should be presented and respectively moved before the results section . In the abstract material and methods are presented at point no 2 before 3 (the results)

Response 2: Thank you for pointing this out. We have now moved the material and methods section as requested, before the result section in the main manuscript.

Comments 3: Reference no 5 is very old and with no direct relation with the manuscript.

Response 3: Reference no. 5 (James et al., 1966) is one of the earliest and the key paper on the human SN anatomy. It is ‘the gold’ reference in understanding the micro-anatomy of the SN. It shows that the SN pacemaker cells are small and lack mitochondria, which aligns with the main idea of that sentence (Lines 39-42 in the introduction).

Comments 4: Some of my comments are inserted directly into the text. Many phrases must be reformulated (from English language point of view)

Response 4: After receiving your comments, Peter Molenaar and Andrew Atkinson (both co-authors of this paper) proofread the revised manuscript one more time. We hope the current revised draft meets your English standards.

Comments 5: In Disscusion section I suggest to present separately the limitations of the study.

Response 5: We have now put limitations in a separate section between the discussion and conclusion (Lines 415-426, highlighted in blue). 

Reviewer 2 Report

Comments and Suggestions for Authors

This report represented the difference in mRNA expression between SN and right atrium.

Authors demonstrated some mRNA such as contractile mRNA, mitochondrial mRNA, and glycogen mRNA was decreased in SN and some TFs and miRs were associated with the decrease of mRNA in SN. There were several issues to be addressed.

# What was the main scientific question in the manuscript? I could not understand the question in introduction.

# In the first place, it seems that each of SN and right atrium is composed of different cell types, and the difference in mRNA is probably due in large part to the difference in cell type. Cell types should be discussed in more detail.

# How about the association between characteristic mRNA profile in SN and SN disease such as sick sinus syndrome?

Comments on the Quality of English Language

No comment.

Author Response

Comment 1: This report represented the difference in mRNA expression between SN and right atrium.

Authors demonstrated some mRNA such as contractile mRNA, mitochondrial mRNA, and glycogen mRNA was decreased in SN and some TFs and miRs were associated with the decrease of mRNA in SN. There were several issues to be addressed.

Response 1: Thank you for reviewing our paper. We carefully address the issues below.

Comment 2: What was the main scientific question in the manuscript? I could not understand the question in introduction.

Response 2: The focus of this paper is to further classify the reduced expression of mRNAs in the SN vs. the right atrium into contractile function related, mitochondrial function related and glycogen metabolism related mRNAs. We combined the key TFs and/or miRs to produce the interaction maps in order to determine any potential novel therapeutic targets to treat SN dysfunction (lines 88-93, highlighted in yellow).

Comment 3: In the first place, it seems that each of SN and right atrium is composed of different cell types, and the difference in mRNA is probably due in large part to the difference in cell type. Cell types should be discussed in more detail.

Response 3: When doing the dissection, only the main body of the SN regions was selected and dissected. We totally agree that the SN is comprised of multiple cell types, such as pacemaker cells, endothelial cells, fibroblasts and macrophages, however, they all come together to maintain the healthy SN function. This is also true for the RA. We tried to only dissect the RA from the pectinate muscles, however, a small amount of the connective tissues and micro-vasculatures can get involved. We added this to the limitation section (lines 423-426 and highlighted in yellow).

Comment 4: How about the association between characteristic mRNA profile in SN and SN disease such as sick sinus syndrome?

Response 4: Thank you for pointing this out. Currently, our molecular profile is based on healthy SN and RA tissues. It would be interesting to do this profile on the SN from diseased and aged hearts and this is what we intend to do in the future.

Although our data from the healthy heart cannot be associated with SN dysfunction directly, we added a paragraph in the discussion section to highlight the potential link of contractile and/or mitochondrial dysfunction with SN dysfunction and atrial fibrillation from some published articles (the references 34-36 and lines 331-336, highlighted in yellow). 

Round 2

Reviewer 2 Report

Comments and Suggestions for Authors

The revised manuscript was finely corrected.

Comments on the Quality of English Language

No comment